# Thermal Performance Improvement by Rotating Thermosyphon Loop in Rotor of an Interior Permanent Magnet Synchronous Electric Motor

**Pey Shey Wu [1]**, **Min-Fu Hsieh [2]**, **Yong En Lu [1]**, **Wei Ling Cai [3]** and **Shyy Woei Chang [3],***

[1] Department of Mechanical and Automation Engineering, Da-Yeh University, No. 168, University Road, Dacun 51591, Taiwan; pswu@mail.dyu.edu.tw (P.S.W.); R1011002@cloud.dyu.edu.tw (Y.E.L.)
[2] Department of Electrical Engineering, National Cheng Kung University, Tainan City 70101, Taiwan; mfhsieh@mail.ncku.edu.tw
[3] Department of System and Naval Mechatronic Engineering, National Cheng Kung University, No. 1, University Road, Tainan City 70101, Taiwan; P18071018@mail.ncku.edu.tw
* Correspondence: swchang@mail.ncku.edu.tw

**Abstract:** As an attempt to enable a further increase in the power-to-weight ratio of an electric motor by improving its cooling performance, rotating thermosyphon loops in a rotor of a permanent magnet synchronous electric motor are proposed. The effective thermal conductivity and airflow heat-transfer rate of the rotating thermosyphon loop and the convective heat-transfer coefficient over the annular interior surface of the air chamber are measured to permit the definition of the thermal boundary conditions for simulating the temperature fields of the electric motors. The axial heat-transfer pathway with extremely high effective thermal conductivity attributing to the phase-change activities in the rotating thermosyphon loop acts synergistically with the heat convection enhancement induced by the stirring effect of the spinning condenser bend in the air chamber to improve the heat transmission out of the rotor core. The spatially average temperature gradients in the rotor with the thermosyphon loops are considerably moderated from those without the thermosyphon loop. At rotor speeds and electrical currents in the ranges of 1200–1500 rev/min and 1000–1200 A, the maximum temperatures in the rotors with the single- and twin-end rotating thermosyphon loops are, respectively, reduced 8–14 °C and 10–22 °C from those without a rotor-cooling scheme, affirming the effectiveness of a phase-change cooling device in a rotor for thermal performance improvement of an electric motor.

**Keywords:** rotating thermosyphon loop; rotor cooling; electric motor

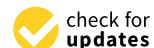



## 1. Introduction

Greenhouse gas emissions cause global climate change and promote green energy development. The green energy sources such as wind, hydro and solar powers are mostly converted into electricity, leading to the gradual popularization of the electric power train. However, the various motor/generator losses, including copper, magnet, and eddy-current losses of an electric machinery converted into thermal powers to raise the temperatures of a motor/generator and deteriorate the electromagnetic efficiency and life span. The efficiency of an electric motor/generator decreased with the temperature increase, owing to the thermal impact on the saturated magnetic flux density [1]. Reducing the temperature of an electric motor/generator is important for improving its efficiency. In particular, an electric motor/generator with large capacity or high power-density requires a cooling system to maintain its operation temperature in the sustainable range, but consumes cooling power, resulting in motors/generators becoming one of the mainstream forms of world energy consumption [2]. In [3], an efficiency improvement of 2% for general motors/generators led to the reduction of energy consumption by about 25%. Moreover, with the promotion of maglev trains and electric vehicles in recent years, electric motors have developed towards

a high power-to-weight ratio and miniaturization, which has strengthened the importance of thermal analysis for an electric motor. Relevant thermal studies have included the electric propulsion powers [4–6], the magnetic–thermal coupling analysis [7–9], and the thermal performances of air- [10–18], water- [19,20], and oil-cooled electric motors [21]. For motors with large power-density, the use of liquid cooling and heat pipes is more effective than that of air-cooled motors [18,22–24]; but cooling systems for low-power-density motors are still dominated by air-cooled motors for low-cost economy [10–18]. The effectiveness of liquid-spray, air, and/or water cooling systems was further enhanced when the motor components were made from materials or compounds with high thermal emissivity and conductivity [25]. The integration of stator and shaft cooling systems enabled a further boost of power density, and the temperature of the motor stator coil was reduced by about 50% and 38%, respectively, compared to the conventional motors and the air-cooled motors; whereas the case temperature of the motor was reduced by 10–42% [26].

Recently, the embodiment of high cooling capacity with a significant heat-transfer enhancement (HTE) was accomplished by using phase-change heat-transfer processes for cooling of the stator [27] and rotor [28–32] of rotating machinery. Putra and Ariantara [27] installed eight L-shaped flat heat-pipes to transfer the heat flux emitted from the rotor to the coolant circuit of the stator. The combination of the water (air) cooling network with the L-shaped heat pipes reduced the casing temperature to 68.4 °C from 102.2 °C at a thermal power of 150 W [27]. Niti et al. [28] configured the pulsating heat pipe into 11, 22, and 33 petal-shaped loops to operate at a centripetal acceleration between 0.5 and 20 g to explore the effect of centrifugal force and the number of turns on its cooling performance. When the rotational speed of the pulsating heat pipe was increased, the thermal resistance kept decreasing. As the flow resistance also increased with the total length of the pulsating heat pipe and the number of turns, the average thermal resistance increased with the number of loops [28]. The subsequent studies explored the effects of input heat flux, rotational speed, filling rate, and working fluid on the cooling performance of the rotational pulsating heat-pipes [29]. In the speed range of 200–800 rev/min, the optimal filling ratio was 50%. Using water or ethanol as the working fluid, the thermal resistance at the optimal condition was reduced by 5.4% and 13% formed that at the static condition [29]. To transmit the thermal power in a rotor to the coolant stream through a hollow shaft of an electric motor, the closed thermosyphon networks were configured into the pad-shaped discs within which the liquid–vapor circulations were motivated by the rotation [30,31]. The thermal impacts of boiling number, relative centrifuge, and condenser thermal resistance on the cooling performances of the rotating pads were studied and casted in the empirical correlations of overall thermal resistances [30,31]. A liquid spray served as an alternative phase-change cooling method for electric motors [32]. Its HTE mechanism was evolved from the phase-change activities after the spray impinging on the hot motor component. When the magnetic field fluctuation or electric noise was moderated using such a liquid-spray cooling scheme, the large pressure drop across the spray nozzle was inevitable to add the cooling power consumption [33].

Along with the development of motor-cooling technology [4–32], the studies of thermal management for an electric motor synchronously progressed [34–37]. The early works in this research field adopted the lumped-parameter thermal network method to model the various motor components into a thermal resistor, thermal capacitor, or heat source [34,35]. Owing to the difficulties in converting the compounded components and the thermal boundary conditions into the equivalent thermal network parameters, the lumped-parameter thermal network method evolved into the coupled thermal electromagnetic analysis [36,37]. This method commenced from the electromagnetic analysis under a set of predefined power output, speed, and constructional details. The flux and leakage distributions of the magnetic and electric fields, the efficiency chart, and the various electromagnetic power losses were numerically determined. The various losses were converted into the volumetric thermal power generations in the power-loss components. Thermal network analysis was then carried out to estimate the average temperatures of the components

with their temperature-dependent thermal properties to be iteratively solved [37]. Since the motor components were modeled as the lumped parameters in a thermal network, the detailed temperature distributions in a component, such as coiled winding and magnet in rotor were not attainable, which prohibited the precise identifications of each hotspot and its maximum temperature level.

When modeling a thermal field of an electric motor, the evaluation of effective specific heat and thermal conductivity for a coiled winding, which is composed of copper wires and thermal insulation paste of different properties, remains a difficult task, especially with the filling ratio effect. Our previous work [38] proposed the non-homogeneous model to take into account the effects of filling ratio and the various material properties when evaluating the effective thermal conductivity and specific heat of a compound motor component such as coiled winding. In this regard, a winding coil is composed of the copper wire in the thermally insulated paste at a specific filling ratio. The effective thermal conductivity of this motor component becomes directional. The thermal resistances formulated by the copper-wire bundle and the insulation paste are thermally connected in shunt and in series along axial and radial directions to result in the maximum and minimum effective thermal conductivity, respectively. The effective thermal conductivity prescribed in a polar coordinate system for a compound coiled-winding [38] is converted to that in a $xyz$ frame of reference to define its directional effective thermal conductivity $k_{eff,x}$, $k_{eff,y}$, and $k_{eff,z}$; especially at each bend of a coiled winding. In order to identify the hotspot location and its temperature level in each motor component based on the thermal powers converted from the electromagnetic losses, the detailed temperature distribution in each component of an electric motor was revealed using the ANSYS commercial code [38]. The axial and radial convective thermal resistances surrounding a rotor, which was shielded by the Taylor vortices in the stator-to-rotor annular gap and the air chambers between the axial ends of rotor and motor casing, undermined the effective heat-transmission from the rotor toward the cylindrical water jacket, leading to hotspot development in a rotor when a shaft cooling scheme was absent [38].

Considering the significance of rotor cooling for the pursuit of ever-mounting power-density of an electric motor, the thermosyphon loops are implanted in the rotor to construct a highly efficient and effective heat-transfer pathway without additional cooling power-consumption. The centrifuge-driven vapor–liquid circulation that axially transfers the thermal power from the rotor to the heat sink (air chamber) in the axial end of the motor utilizes the highly efficient phase-change heat-transfer processes. Figure 1 depicts the constructions of the electric motors (a) without and with (b) single-end (c) twin-end rotating thermosyphon loops (RTLs) in the rotor. With five (ten) RTLs in the rotor, the length and diameter of each RTL are 156 mm (102 mm) and 4 mm (4 mm) with 17% of total length exposing in the air chamber as the condenser bend. For the twin-end RTL arrangement, the overlapped axial length in the rotor between the RTLs inserted from two axial ends of the rotor is 5% (14.7%) of axial length of the rotor (total RTL length). In the rotor, the vapor generated in the evaporator of each RTL is drawn axially toward the cold condenser bend where the condensates are formed and impelled forward the radially outer thermosyphon tube. The heat flux from the rotor to the air chamber(s) at the axial end(s) of the electric motor is transferred toward the coolant stream in the multi-pass serpentine channel of the cylindrical water jacket. The novelty of this invention lies in the utilization of the effective and efficient phase-change heat-transfer activities to axially transport the thermal power from the rotor core to the air chamber(s) underneath the cylindrical water jacket with the attendant reduction of the convective thermal resistance in the air chamber by stirring the airflow with the spinning 180° tube bends of the RTLs. The present study is formulated to investigate the comparative thermal performances between the electric motors without and with the single-and twin-end rotating thermosyphon loops in the rotor. The research goal is fulfilled by completing the measurements of the effective thermal conductivity of the RTL and the convective heat-transfer rate surrounding its condenser bend to permit the emulation of the interior thermal boundary conditions for simulating the

detailed temperature distributions of the electric motors. The integration of the numerical and experimental methods developed by this research group [38,39] is extended to newly explore the impact of the in-rotor RTLs on the thermal performance improvement of the permanent magnet synchronous electric motor (PMSEM). In what follows, the incorporation of the experimental findings [39] and the thermal–electromagnetic model [38] is briefly described with a detailed illustration for the transient heat-transfer measurement technique for acquiring the Nusselt numbers on the annular surface of the air chamber with the RTLs. The subsequent analysis of the thermal fields in the PMSEMs with and without the RTLs is carried out numerically in an attempt to highlight the practical significance of a phase-change cooling-device on the thermal performance improvement for an electric motor.

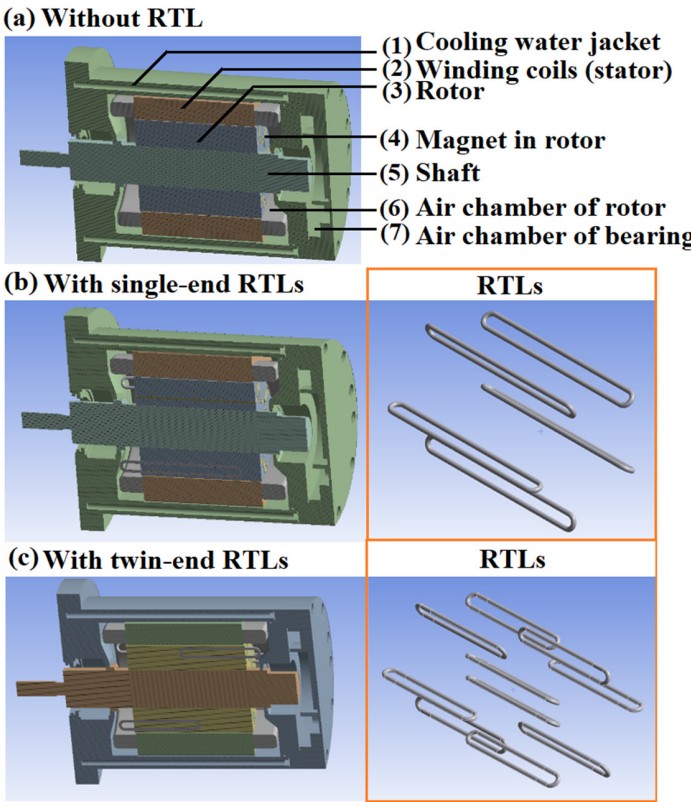

**Figure 1.** Constructions of the electric motors (**a**) without and with (**b**) single-end (**c**) twin-end thermosyphon loops in the rotor.

The research goal aimed at utilizing the phase-change activities with extremely high heat-transfer rates and free of cooling power-consumption puts forward the advancement of current motor rotor cooling technology. To fulfill the reasserted goal numerically, the heat-transfer parameters involved in the numerical model with RTLs in the rotor are essential. Referring to Figure 1, the thermal impact of the complex boiling and condensation activities in the rotating thermosyphon loop is quantified as the effective thermal conductivity ($k_{eff}$) of the RTL in the numerical model. In [39], $k_{eff}$ is the function of dimensionless thermal power and centripetal acceleration of the RTL. As the thermal power converted from the rotor losses is efficiently and effectively transferred to the annular air chambers at the front and rear sides of the rotor via the RTLs, the average Nusselt number on the outer surface of the condenser bend of the RTL ($Nu_{ext,con}$) that rotates in the stirred airflow field of each air chamber, as well as the Nusselt number on the inner annular surface of the air chamber ($Nu_{inner,AC}$) are the necessities for defining the internal thermal boundary conditions in the numerical model. As the airflow field is induced by rotation, both the $Nu_{ext,con}$ and $Nu_{inner,AC}$ increase with the rotating speed. In addition to the thermal parameters associated with the RTLs in the numerical model, the various heat-convective conditions are inherited from the shaft-driven rotor-rotation in the stator [38]. In this respect, the Nusselt number

of Taylor vortical flow ($Nu_{Taylor}$) in the annular stator-to-rotor gap, the rotating-disc-like Nusselt number on the two axial planar ends of rotor ($\overline{Nu_B}$), and the Nusselt number on the outer surface of the rotating shaft ($\overline{Nu_C}$) are the rotation-dependent flow parameters. Inside the present PMSEM, the free convective Nusselt numbers over a vertical plate ($Nu_{VPW}$) and around the inner surface of a horizontal cylinder ($\overline{Nu_{HC}}$) are evaluated using the well known free convective heat-transfer correlations [38]. The determinations of the aforementioned convective boundary conditions in terms of the various Nusselt numbers are illustrated along with the associated experimental methods in the following section.

## 2. Methods

### 2.1. Experimental Method

#### 2.1.1. Steady State Thermocouple Method

The effective thermal conductivity and the external Nusselt number of the rotating 180° condenser bend of the RTL are measured by the steady-state thermocouple method, which has been reported in [39]. The Nusselt numbers on the annular inner surface of the cylindrical air chamber, within which the extended 180° condenser bends rotate with the rotor assembly, are measured by the newly devised transient infrared thermography method. This section briefly describes the steady-state thermocouple method [39] with a detailed illustration of the transient infrared thermography method.

Figure 2 depicts (a) rotating rig that carries (b) RTL at a controllable speed in the range of 100–400 rev/min with the eccentricity of 450 mm, giving rise to the maximum centripetal acceleration of 66.62 g. The rotating rig shown by Figure 2a is driven by a DC motor. The electrical heating power from an adjustable DC supply to RTL is transmitted through the copper slip-ring unit. All the temperature and pressure signals collected from the RTL are connected with the Fluke NetDAQ data logger via a 36-channel instrumentation slip ring. An online condition monitoring program scans the instant RLT temperatures. A steady state is reached when the discrepancy between a series of successive wall-temperature ($T_w$) scans is less than 0.3 °C. Having satisfied the steady-state condition, all the raw data are stored in the online data-acquisition program for the subsequent analysis.

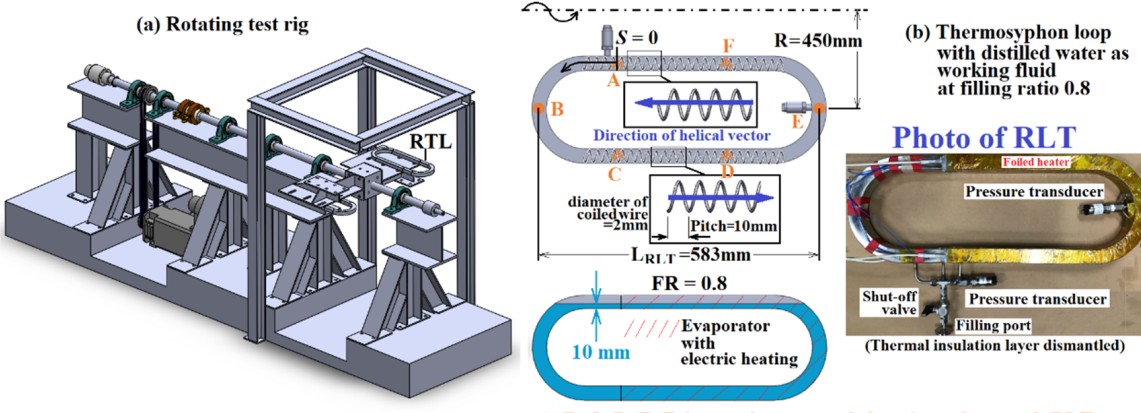

**Figure 2.** (**a**) rotating test rig (**b**) RTL for measuring the effective thermal conductivity and the external Nusselt number of the 180° condenser bend of the RTL using the steady-state thermocouple method [39].

The RTL is made from a 1 mm thick stainless-steel square duct with an inner width/height of 30 mm. The channel hydraulic diameter (*d*) of 30 mm is selected as the characteristic length for processing this set of heat-transfer data. The heating foils are mounted on the outer surfaces of the evaporator section, as indicated in Figure 2b. A thin layer of thermal paste is applied between the heating foil and the RTL outer wall. The axial length from the tip of the evaporator (the condenser) bend to the edge of the heating foil at the junction between the evaporator and the condenser sections is 410 (203) mm. The

nominal length ($L_{RTL}$) and the width between the inner and the outer legs of the RLT are 583 and 190 mm. The centerline curvature of the 180° evaporator or condenser bend is 80 mm. There are three thermocouples and one pressure transducer installed in the evaporator or the condenser section. The thermocouple location is 0.5 mm away from the inner wall of the RTL. Having acquired the local heat flux transferred in and out of the evaporator and the condenser, the measured wall temperature ($T_w$) is corrected to that at the inner duct wall using one-dimensional Fourier conduction law. The evaporator section of the RTL is wrapped by a thick thermal insulation layer to reduce the external heat loss. The airflow effect on the $T_w$ measurements over the condenser section is minimized by covering a spotted thermal insulation pad on each foil-type thermocouple. The evaporator and the condenser pressures of the RTL are measured by the piezo-metric-type pressure transducers at the central of evaporator bend and the inner leg of the condenser section, as indicated in Figure 2b. The RTL is filled with degassed and distilled water with a volumetric filling ratio of 0.8 at the absolute pressure of 7.13 Nm$^{-2}$. At the filling ratio of 0.8, a 10 mm thick liquid film immerses the inner sidewall of the rotating thermosyphon loop to ensure the boiling activities in the evaporator of RTL at all the rotating teat conditions.

To feature the complex vapor–liquid phase-change heat-transfer process in the RTL as a conductive pathway in a rotor of an electric motor, the effective thermal conductivity ($k_{eff}$) of the RTL at a set of angular velocity ($\Omega$) and thermal power ($Q$) is measured by the steady-state heat-transfer measurement method using Equation (1).

$$k_{eff} = Q / \left[ \left( \overline{T_w}_{\text{eva}} - \overline{T_w}_{\text{con}} \right) \times L_{RTL} \right] \tag{1}$$

As previously illustrated, the averaged wall temperatures of the evaporator ($\overline{T_w}_{\text{eva}}$) and the condenser ($\overline{T_w}_{\text{con}}$) in Equation (1) are averaged from the measured wall temperatures along the RTL. The thermal power fed to the evaporator of the RTL ($Q$) is determined after subtracting the external convective heat-loss power from the supplied electrical heating power. The external convective heat-loss power is correlated into the function of ($\overline{T_w}_{\text{eva}} - T_\infty$) and $\Omega$ via a series of heat-loss calibration tests, as reported in [39]. Through the parametric analysis for correlation development, the non-dimensional effective thermal conductivity of the RTL ($K_{eff}$) is devised as the function of dimensionless thermal power ($Q^*$) and relative centripetal acceleration ($Ca$) using the following groups:

$$K_{eff} = k_{eff} / k_w \tag{2}$$

$$Q^* = (Qd) / (\mu_f h_{fg}) \tag{3}$$

$$Ca = \Omega^2 R / g \tag{4}$$

In Equations (2)–(4), $k_w$, $d$, $\mu_f$, $h_{fg}$, $\Omega$, $R$, and $g$, respectively, stand for the thermal conductivity of the RTL duct wall, the hydraulic diameter of the RTL duct, the liquid-phase viscosity and the latent heat of working fluid in RTL, the angular velocity, the rotating radius of RTL, and the gravitational acceleration.

In Figure 1, the extension of the 180° condenser bend of each RTL into the air chamber at the front/back end of the rotor allows the rotation-induced airflow to convect the thermal power ($Q$) into the air chamber. Another important convective performance factor of the RTL for a simulation of the thermal filed in an electric motor is the average heat convection rate over the external surface of the rotating condenser bend. The steady-state heat-transfer tests using the test facilities of Figure 2 generate the required data, since the convective heat flux transferred into the condenser ($q_{con}$), which is equivalent to the heat flux of evaporator ($q_{eva}$) at a steady state condition, is deemed to be balanced with the convective heat flux carried away by the external airflow over the rotating 180° condenser bend at an air temperature of $T_\infty$. The average Nusselt number of the airflow surrounding the rotating condenser bend of the RTL ($Nu_{ext,con}$) is measured by Equation (5).

$$Nu_{ext,con} = \left[ q_{con} / \left( \overline{T_w}_{\text{con}} - T_\infty \right) \right] d / k_{air} = h_{ext,con} d / k_{air} \tag{5}$$

In Equation (5), $k_{air}$ is the thermal conductivity of air at the measured ambient temperature ($T_\infty$) [39]. In [39], the $Nu_{ext,con}$ data are governed by the rotating Reynolds number ($Re_\Omega$) taking the dimensionless form of:

$$Re_\Omega = \Omega R^2 / \nu \tag{6}$$

In Equation (6), R is the rotating radius of RTL. The kinematic viscosity of the airflow ($\nu$) in Equation (6) is also evaluated at the ambient temperature ($T_\infty$). In the electromagnet thermal model, the convective heat-transfer coefficient surrounding each 180° condenser bend of the RTL ($h_{ext,con}$) is evaluated from the $Nu_{ext,con}$ correlation generated at this phase of experimental study with $T_\infty$ referring to the gas temperature ($T_g$) of the air chamber. The determination of $T_g$ in the air chamber via the iterative scheme for simulating the thermal field of an electric model will be later illustrated.

### 2.1.2. Transient Thermography Method

In additional to the relevant Nusselt number correlations involving rotating surface or free convection for thermal simulation of an electric motor [38], it is crucial to identify the heat-transfer impact of the rotating 180° condenser bend on the inner annular surface of the air chamber for closing the definition of the interior thermal boundary conditions when simulating the thermal field of an electric motor with the RTLs in the rotor. In this regard, the airflow in the enclosed air chamber is established by the viscous entrainment of the rotating rotor assembly with or without the stirring flow induced by the spinning condenser bends. Such a flow characteristic prohibits the implementation of a steady-state heat-transfer measurement method for acquiring the Nusselt number data over the inner annular surface of the air chamber ($Nu_{inner,AC}$). Alternatively, the $Nu_{inner,AC}$ data is acquired by applying the transient infrared thermography measurement method invented by the present study. Figure 3 depicts the test rig for detecting the distribution of $Nu_{inner,AC}$ over the inner annular surface of the cylindrical air chamber with and without the RTLs using the transient infrared thermography method. The geometric characterization of the air chamber on each axial end of the test rig complies with the interior configurations of the PMSEM reported in [38].

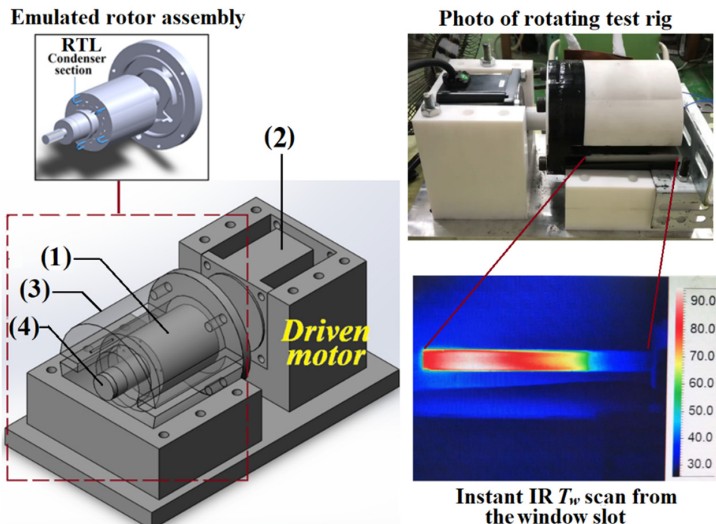

**Figure 3.** Test rig for measuring $Nu_{inner,AC}$ map on the annular inner surface of the cylindrical air chamber with and without the RTLs.

As shown by the photo of the test rig in Figure 3, the rotor assembly, the thermal insulation shields around the stainless-steel annular heating foil, and the foundation of the test rig are made of Teflon. The Teflon rotor assembly (1) is driven by a DC electric motor (2). A 0.1 mm thick stainless steel annular foil is mounted underneath the cylindrical Teflon

shield (3). The end bearing supports the shaft (4) that rotates at a controllable speed. The airflows in the air chamber and the interconnecting annular gap between the rotor and the stainless-steel annular foil, which emulates the inner surface of the stator, are induced by the rotating motion of the rotor assembly with and without the five or ten RTL bends. To perform each transient heat-transfer test, the electrical heating power is initially fed to the 0.1 mm thick stainless-steel annular foil for raising the gas temperature ($T_g$) in the air chamber. Five thermocouples with an equal angular interval penetrate into the air chamber to measure the air temperatures. The average of these five thermocouple readings is treated as the instant $T_g$ in the air chamber. A rectangular slot through the cylindrical Teflon shield (3) permits the instant $T_w$ scan by the infrared radiometer (IR). The present IR (Ching Hsing Computer-Tech Ltd. model P384A3–20) is capable of scanning a $T_w$ map at the rate of 60 fps with a maximum uncertainty of 0.4 °C. The ratio between the arc width of the slot and the projected straight width of the slot is 1.006, which is similar to a flat facet in parallel with the lens of IR. To enhance the radiative emissivity, the back surface of the stainless-steel annular foil is sprayed by black paint. The radiative emissivity ($\varepsilon$) of the black paint is 0.96. As exemplified by Figure 3, the instant $T_w$ image taken from the 0.1 mm thick stainless-steel annular foil through the rectangular slotted window is subjected to the considerable axial gradients with angular variations when the rotor spins. The Biot numbers for all the heat-transfer data generated in this phase of study are less than 0.05 so that the scanned $T_w$ map from the back of the scanned foil is treated as the inner $T_w$ temperatures of the stainless-steel annular foil.

The heat-transfer measurements over the inner annular surface of the cylindrical air chamber are carried out using the rotor assemblies with and without the RTLs at the rotating speeds of 500, 1000, 1500, 2000, 2500, and 3000 rev/min. At each rotor speed tested, the electrical power fed to the stainless-steel annular foil converts directly to the Joule thermal power to raise $T_w$ and $T_g$. The temporal variations of local $T_w$ and $T_g$ follow the general asymptotic exponential-like increases toward their steady-state levels. Having reached the steady-state condition, the electrical power fed to the stainless-steel annular foil is shut off and the $T_g$ is initially raised and then reduced temporally. During the transient period of $dt$ that $T_g$ is temporally reduced, the energy-conservation scenario for a small element $dxrd\theta t$ taken from the scanned stainless-steel foil is featured by Figure 4.

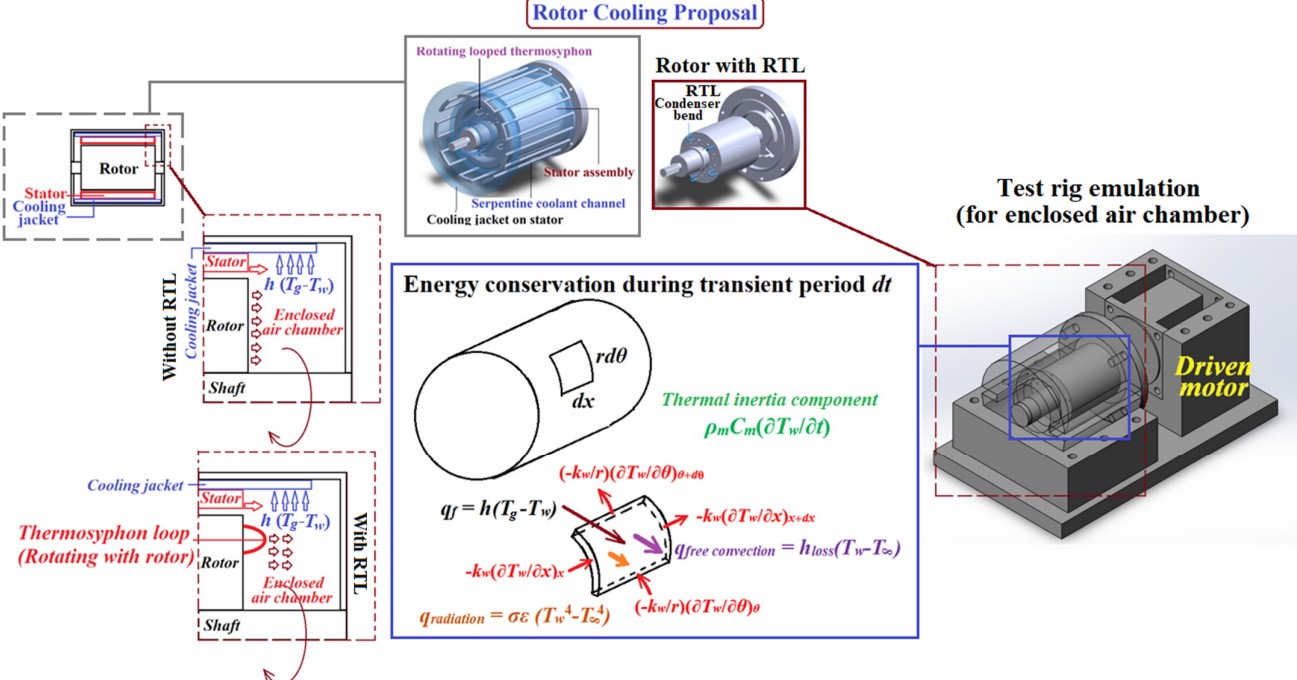

**Figure 4.** Energy conservation scenario of a small element $dxrd\theta$ taken from the stainless-steel foil confined within the slotted window during the transient period $\delta t$.

As shown by Figure 4 that also clarifies the geometric similarity between the present test model and the simulated PMSEM, the heat-transfer scenario for the small element $dxrd\theta$ with a thickness $t$ and thermal conductivity of $k_w$ is expressed by Equation (7).

$$h_{inner,AC}(T_g - T_\infty) = \sigma\varepsilon\left(T_w^4 - T_\infty^4\right) + h_{\text{free-convection}}(T_w - T_\infty) - k_w t\nabla^2 T_w + \rho_m C_m \frac{\partial T_w}{\partial t} \qquad (7)$$

In Equation (7), the local convective heat flux ($h_{inner,AC}(T_g - T_\infty)$) transferred from the hot air at temperature $T_g$ into the element $dxrd\theta t$ at the heat-transfer rate of $h_{inner,AC}$ is balanced with the external radiative ($\sigma\varepsilon(T_w^4 - T_\infty^4)$) and free-convective ($h_{\text{free-convection}}(T_w - T_\infty)$) heat loss fluxes, the conductive heat flux ($k_w t\nabla^2 T_w$) prevailing within the element $dxrd\theta t$, and the thermal inertia of the element ($\rho_m C_m \frac{\partial T_w}{\partial t}$) at each instant. The first three components on the right-hand side of Equation (7) are evaluated from an instant $T_w$ map scanned by the IR at an ambient temperature $T_\infty$. The value of $h_{\text{free-convection}}$ is obtained by substituting the measured local $T_w$ and $T_\infty$ into the Nusselt number correlation [40] for a free convective flow around a horizontal cylinder with a diameter D as Equation (8) for the laminar flow with $10^{-6} < GrPr < 10^9$.

$$Nu_{\text{free-convection}} = h_{\text{free-convection}}D/k_{\text{air}} = 0.36 + \frac{0.518(GrPr)^{1/4}}{\left[1 + (0.559/Pr)^{9/16}\right]^{4/9}} \qquad (8)$$

The thermal expansion coefficient ($\beta$) and the kinematic viscosity ($v$) of air in Equation (9) for calculating local Grashof numbers ($Gr$) are calculated at the local film temperature of $(T_w + T_\infty)/2$.

$$Gr = [g\beta(T_w - T_\infty)]D^3/v^2 \qquad (9)$$

Owing to the fluctuations of the fluid temperatures in the air chamber with a rotating rotor and/or RTL, the temporal $T_w$ variations are frozen as the spatial small-scale $T_w$ fluctuations in each instant $T_w$ map to prohibit the direct numerical differentiation for the accountancy of local $\nabla^2 T_w$ in the third term on the right-hand side of Equation (7). The mean-averaged filtration scheme is therefore adopted prior to the numerical differentiation for unraveling the distribution of $k_w t\nabla^2 T_w$, or the conductive heat flux, on the scanned foil. The final thermal inertial term in Equation (7) is evaluated from two successive filtered $T_w$ maps with a lapse of d$t$. The temporal $T_w$ derivative is multiplied by the specific heat ($C_m$) and density ($\rho_m$) of the stainless-steel foil to determine the distribution of $\rho_m C_m \frac{\partial T_w}{\partial t}$ over the scanned area. As an illustrative example, Figure 5 depicts a set of $T_w$, $\sigma\varepsilon(T_w^4 - T_\infty^4)$, $h_{\text{free-convection}}(T_w - T_\infty)$, $k_w t\nabla^2 T_w$, and $\rho_m C_m \frac{\partial T_w}{\partial t}$ leading to the generation of $h_{inner,AC}$ map over the scanned inner annular surface of the air chamber within which five or ten 180° condenser bends rotate at 2000 rev/min. While the distribution of $h_{inner,AC}$ over the inner annular surface of the air chamber is not uniform in Figure 5, the averaged Nusselt number ($Nu_{inner,AC}$) over the inner annular surface of the air chamber defined as ($h_{inner,AC}$ D)/$k_{\text{air}}$ at each $Re_\Omega$ is evaluated from the averaged $h_{inner,AC}$.

The estimation of experimental uncertainties of the measured dimensionless flow parameters in the present experimental program follows the statistical inference of Kline and McClintock [41]. For the steady-state heat-transfer data, the instrumental precisions, data ranges and maximum error percentages of the measurements were previously reported with the maximum uncertainties of $Ca$, $Q^*$, $Re_\Omega$, $K_{eff}$, and $Nu_{ext,con}$ [39] collected in Table 1. For the data generated by the transient thermography method, the error sources of $T_g$, $T_\infty$, and $T_w$ were inherited from the thermocouple and IR measurements with the respective uncertainties of 0.3 °C and 0.4 °C according to their calibration reports. With $T_w$, $T_g$, and $T_\infty$ in the respective ranges of 331–354 °C, 339–360 °C, and 298–300 °C, their maximum error percentages were 0.12%, 0.008%, and 0.02%. Regarding the $Re_\Omega$ accountancy, the precision of 0.1 rev/min for the rotating speed detector gave rise to the maximum error percentage

of 0.02% in the speed range of 500–3000 rev/min. As the fluid properties were evaluated by the equations correlated from the table values with the fluid temperature as the dependent variable, the maximum discrepancies between the correlated and table values of $k_{air}$ and $v$ were $\pm 0.03\%$ and $\pm 0.02\%$, which were negligible. With the manufacturing tolerance of $\pm 0.01$mm for the 82.2 mm inner diameter of the cylindrical air chamber, the percentage error of D was 0.012%. After performing the error propagation analysis, the maximum root mean square experimental uncertainties at 95% confidence interval for $Re_\Omega$, $Gr$, and $Nu_{inner,AC}$ were estimated by the present study as indicated in Table 1.

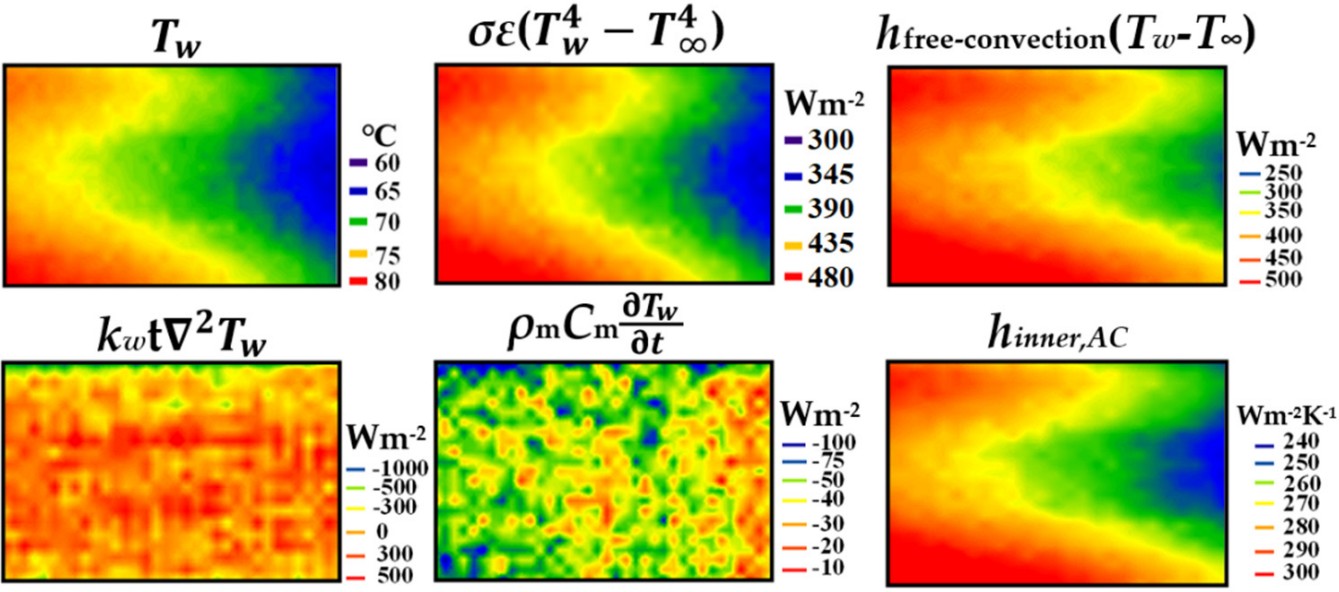

**Figure 5.** An illustrative set of $T_w$, $\sigma\varepsilon(T_w^4 - T_\infty^4)$, $h_{\text{free-convection}}(T_w - T_\infty)$, $k_w t\nabla^2 T_w$, and $\rho_m C_m \frac{\partial T_w}{\partial t}$ leading to generation of $h$ map over the scanned inner annular surface of the air chamber, within which five 180° condenser bends rotate at 2000 rev/min.

**Table 1.** Experimental uncertainties of the measurements.

| Experimental Uncertainties for RTLs [39] | | | | | |
|---|---|---|---|---|---|
| Parameter | $Ca$ | $Q^*$ | $Re_\Omega$ | $K_{eff}$ | $Nu_{ext,con}$ |
| Uncertainty (%) | 1 | 7.42 | 1.56 | 8.9 | 1.57 |
| **Experimental Uncertainties for Measuring Nusselt Number on Annular Inner Surface of Air Chamber with RTLs** | | | | | |
| Parameter | $Re_\Omega$ | | $Gr$ | $Nu_{inner,AC}$ | |
| Uncertainty (%) | 0.8 | | 8.42 | 8.2 | |

### 2.2. Numerical Method

The numerical simulation utilizes the electromagnetic thermal model reported previously in [38]. The grid structure and the various losses (thermal powers) generated by the motor components at the three simulation conditions, the convective thermal boundary conditions of the electric motor, the iterative procedure for determining $T_g$ in each air chamber, and the model validation are illustrated as follows.

Figure 6 depicts (a) grid structure, (b) flow regions that require the definitions of convective thermal boundary conditions in the electric motor, and (c) the grid dependency test result. For the thermal field analysis, three motor configurations as shown as Figure 1 were analyzed under three sets of identical operating conditions referred to as $T_0$, $T_1$, and $T_2$ cases for the motors with and without the RTL. The rotor iron loss, stator iron loss, PM loss, and copper loss at the $T_0$–$T_2$ operating conditions with the specific rotor speeds

and total electric currents fed to the motor were evaluated by the electromagnetic model reported in [38] as Table 2.

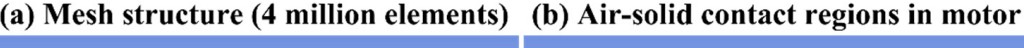

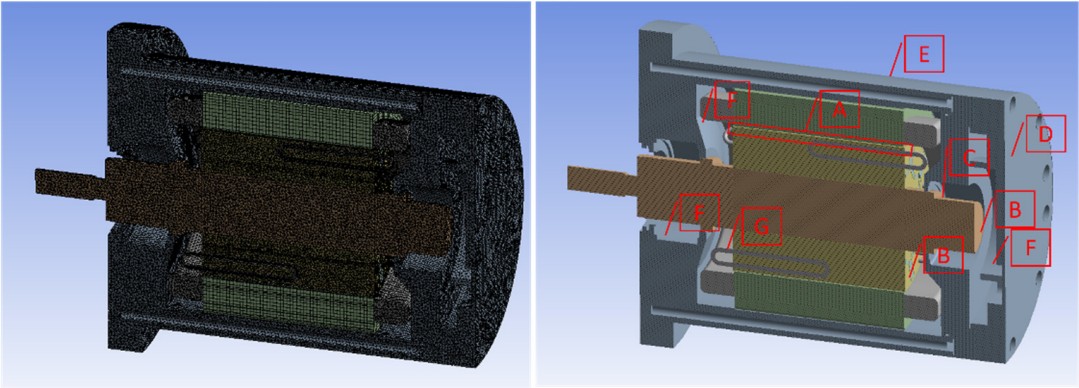

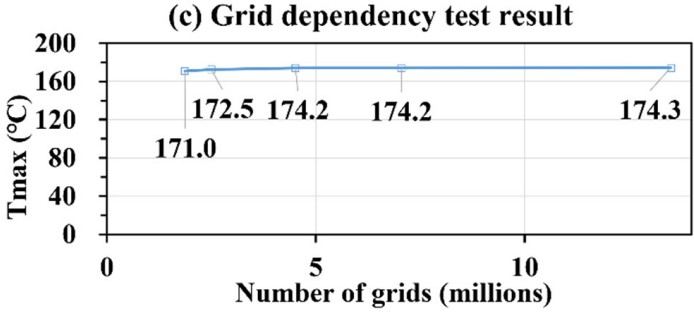

**Figure 6.** (**a**) grid structure; (**b**) convective flow regions in motor; (**c**) grid dependency test result.

**Table 2.** Rotor iron loss, stator iron loss, PM loss, and copper loss at the operating conditions $T_0$–$T_1$ with the specific rotor speeds and total electric currents.

|  | Rotor Speed (rpm) | Total Current (A) | Rotor Iron Loss (W) | Stator Iron Loss (W) | PM Loss (W) | Copper Loss (W) |
|---|---|---|---|---|---|---|
| $T_0$ | 1200 | 1000 | 60.5 | 101.8 | 188.1 | 4500 |
| $T_1$ | 1500 | 1000 | 87.3 | 138.8 | 296.7 | 4500 |
| $T_2$ | 1200 | 1100 | 65.8 | 102.6 | 224.0 | 5445 |

Case $T_0$ in Table 2 is selected as the baseline reference condition. Cross examinations of the thermal fields obtained at $T_0$ and $T_1$ cases ($T_0$ and $T_2$ cases) unravel the effect of rotor speed (power output) on the cooling performance improvement attributed to the RTLs at a fixed power output (rotor speed).

The geometric characterization of the simulated PMSEM follows that described in [38] with the meshes re-generated to accommodate the RTLs in the rotor and the different water jacket within which the spiral channel [38] is replaced by the serpentine coolant channel. As exemplified by Figure 6a, the structured meshes for the coiled windings in stator and the unstructured meshes for the other motor components construct the present grid system. The aluminum cylindrical water jacket is mounted on the outer surface of the stator. The convective heat-transfer rate of the water flow in the serpentine coolant channel is evaluated by Dittus–Boelter correlation at the Reynolds number of 16,120 for all the simulation cases with and without RTL at $T_0$–$T_2$ operating conditions. Within the electric motor where the entrapped air is in touch with the static or moving surface(s) of the motor components, the convective thermal boundary conditions vary with the flow regions specified in Figure 6b. In addition to the convective boundary conditions in A–E flow regions of Figure 6b which have been previously illustrated [38], the heat convective coefficients on the annular surface of the air chamber with and without the RTLs in the F

region and over the outer surface of the condenser bend as the G region in Figure 6b are required to complete the definition of the thermal boundary conditions. For completeness, all the heat-transfer correlations for the A g flow regions in Figure 6a are summarized in Equations (10)–(17).

In flow region A, the Taylor vortical flow cells are induced by the rotating rotor in the stator with the corresponding Nusselt number ($Nu_{Taylor}$) varying with rotor speed as:

$$Nu_{Taylor} = h_{Taylor}(\text{R}_\text{o} - \text{R}_\text{i})/k_{air} = 0.1548 \times Ta^{0.3039} \tag{10}$$

In Equation (10), the Taylor number (*Ta*) is defined as $[\Omega^2 \text{R}_\text{i}(\text{R}_\text{o} - \text{R}_\text{i})^3]/v^2$, where $\text{R}_\text{i}$ and $\text{R}_\text{o}$ are referred to as the inner radius of the stator and the outer radius of the rotor.

In flow region B, where the rotor with a diameter D rotates at an angular velocity $\Omega$, the average Nusselt number on the rotating planar end facet ($\overline{Nu_B}$) is defined by Equation (11) when the rotating Reynolds number ($Re_\Omega = \Omega \text{D}^2/v$) is less than $10^6$.

$$\overline{Nu_B} = \overline{h_B}(\text{D})/k_{air} = 0.36 \times Re_\Omega^{0.5} \tag{11}$$

In flow region C, the Nusselt number ($\overline{Nu_C}$) over the outer surface of the rotating shaft with a diameter $d_s$ is evaluated by Equation (12) as:

$$\overline{Nu_C} = \overline{h_C}(d_s)/k_{air} = 0.11 \times \left(0.5 Re_\Omega^2 + Gr_{d_s} Pr\right)^{0.35} \tag{12}$$

In Equation (12), the characteristic length of $\overline{Nu_C}$, $Gr_{d_s}$, and $Re_\Omega$ is the shaft diameter ($d_s$).

With the free convective flows over the vertical planar wall and on the annular outer surface of a horizontal cylinder with the characteristic lengths L and D, respectively, the Nusselt numbers $Nu_{VPW}$ and $\overline{Nu_{HC}}$ in flow regions D and F of Figure 6b increased with the Raleigh number ($Ra_{\text{L,D}}$) in according to Equations (13) and (14).

$$Nu_{VPW} = h_{VPW}\text{L}/k_{air} = 0.68 + \frac{0.67 Ra_\text{L}^{0.25}}{\left[1 + (1 + 0.492/Pr)^{9/16}\right]^{4/9}} \tag{13}$$

$$\overline{Nu_{HC}} = \overline{h_{HC}}(\text{D})/k_{\text{air}} = \text{C} \times Ra_\text{D}^n \left(\text{C} = 0.48, \; n = 0.25 \text{ when } 10^4 < Ra_\text{D} < 10^7\right) \tag{14}$$

Regarding the $h_{inner,AC}$ over the annular surface of the air chamber with or without the RTLs in region F, and on the outer surface of the rotating condenser bend in region G, the heat-transfer correlations are, respectively, generated by the transient thermography method of the present study and the steady-state thermocouple method reported in [39]. The details of the Nusselt number correlations expressed by the following Equations (15)–(17) will be illustrated later.

$$Nu_{inner,AC} = h_{inner,AC}\text{D}/k_{air} = 1137 - 918.8 \times \exp(-1.59 \times 10^{-5} \times Re_\Omega) \text{ without RTL} \tag{15}$$

$$Nu_{inner,AC} = h_{inner,AC}\text{D}/k_{air} = 1347.3 - 956.6 \times \exp(-2.29 \times 10^{-5} \times Re_\Omega) \text{ with RTL} \tag{16}$$

$$h_{ext,con} = Nu_{ext,con}k_{air}/d = 7.4767 \times (\Omega\text{R}) + 75.178 \tag{17}$$

In Equations (15) and (16), the characteristic length of $Re_\Omega$ is the radius of the rotor (R), while the characteristic length of $Nu_{inner,AC}$ is the diameter of the annular chamber.

The grid dependency test was carried out at $T_2$ condition using the simulation model with the twin-end RTLs. The variation of maximum material temperature in the motor ($T_{max}$) against the number of elements shown by Figure 6c reveals that the $T_{max}$ discrepancies predicted by the models with an element number exceeding four million is less than 0.06%. The mesh constructed by four million of elements as exemplified in Figure 6a has

reached the grid independency condition. For the electric motors without RTL and with the single-end RTLs, the grid structures and the numbers of element that meet the grid independency condition are similar to the model seen in Figure 6c with the twin-end RTLs.

On the surface of a motor component that contacts air, the conductive heat flux out of the component in the direction normal to its surface ($n$) with a thermal conductivity ($k_w$) and surface temperature ($T$) is equal to the convective heat flux transferred into the air ($h(T - T_g)$). In addition to the various convective heat-transfer rates over the component surfaces ($h$) that are in touch with the entrapped air in an electric motor, the air temperature ($T_g$) is required to close the definition of the interfacial thermal boundary condition specified by Equation (18).

$$h(T - T_g) = -k_w(\partial T / \partial n) \tag{18}$$

Initially, the thermal field of an electric motor was solved at the predefined air temperatures ($T_g$) in each air chamber in the motor. A temporary converged temperature field of the electric motor was acquired to permit the successive evaluation of the air temperature ($T_g$) in an air chamber. As exemplified by Figure 7, which depicts an air chamber enclosed by the various components with different surface area ($A_i$) and convective heat-transfer rate ($h_i$) at the surface temperature ($T_i$), the air temperature ($T_g$) is calculated from an energy balance for the enclosed air using Equation (19), as follows:

$$T_g = \sum h_i A_i T_i / \sum h_i A_i \tag{19}$$

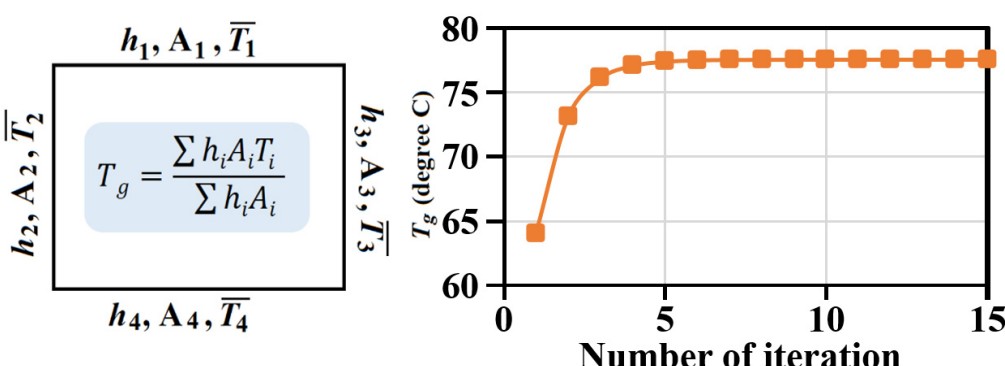

**Figure 7.** Iterative $T_g$ in an air chamber enclosed by various components with different surface area ($A_i$) and convective heat-transfer rate ($h_i$) at the surface temperature ($T_i$).

An iterative scheme performing the ANSYS code with the updated $T_g$ has led the air temperature in each chamber toward a convergent solution, as typified in Figure 7. The numerical solution obtained at the convergence criterion with iterative residue of less than 1% was obtained at $T_0$–$T_2$ simulation conditions using the converged $T_g$ and the convective heat-transfer rates evaluated by Equations (10)–(17).

The verification of the simulation results was carried out by comparing with the experimental temperature data taken from the PMSEM. Figure 8 depicts (a) configuration of electric motor simulated and tested (b) experimental rig and test condition for validation of the simulation model without RTL in rotor (c) comparisons between simulation results and experimental measurements. In compliance with the test facilities, the water flow in the serpentine coolant channel of the jacket around the stator was replaced by air. As indicated in Figure 8b, three thermocouples were attached on the casing with one thermocouple embedded in the stator on the coiled winding to detect the temperature data for model validation. As the alternating magnetic field of the PMSEM affected a thermocouple signal, the shielded thermocouple was located outside the magnetic field established between the rotor and stator at the front face of the coiled winding in the PMSEM. The transient temperature measurements for a period of 1592 s at a rotor speed of 2000 rev/min with a set of nominal electric current and voltage of 60A and 150V were recorded. At this test

condition, the various losses in terms of volumetric thermal power sources in the PMSEM were evaluated as shown in Figure 8b. During the transient period, the iterative $T_g$ scheme is inclusive in the transient simulation model with the temperature dependent convective thermal boundary conditions and fluid properties evaluated at the converged instant thermal field. As shown by temperature snapshot at 1592 s after feeding the electric power into the motor, the casing temperature map and the internal thermal field are subjected to considerable axial and radial variations. On the casing, the discrepancies between the measured and simulated transient variations of the averaged casing temperature were less than 2.3% as compared in Figure 8c. Regarding the interior thermal field, the simulated temporal temperature variation at the front facet of the winding coil in the stator agreed favorably with the measured data as compared in Figure 8c. The maximum discrepancy between the simulated and measured temperatures at the winding coil in the stator was less than 1.7%. The agreements between the measured and simulated temperatures on the casing and in the rotor favorably validated the present simulation model to permit the numerical exploration of the RTL effect on the thermal performance improvements for the PMSEM.

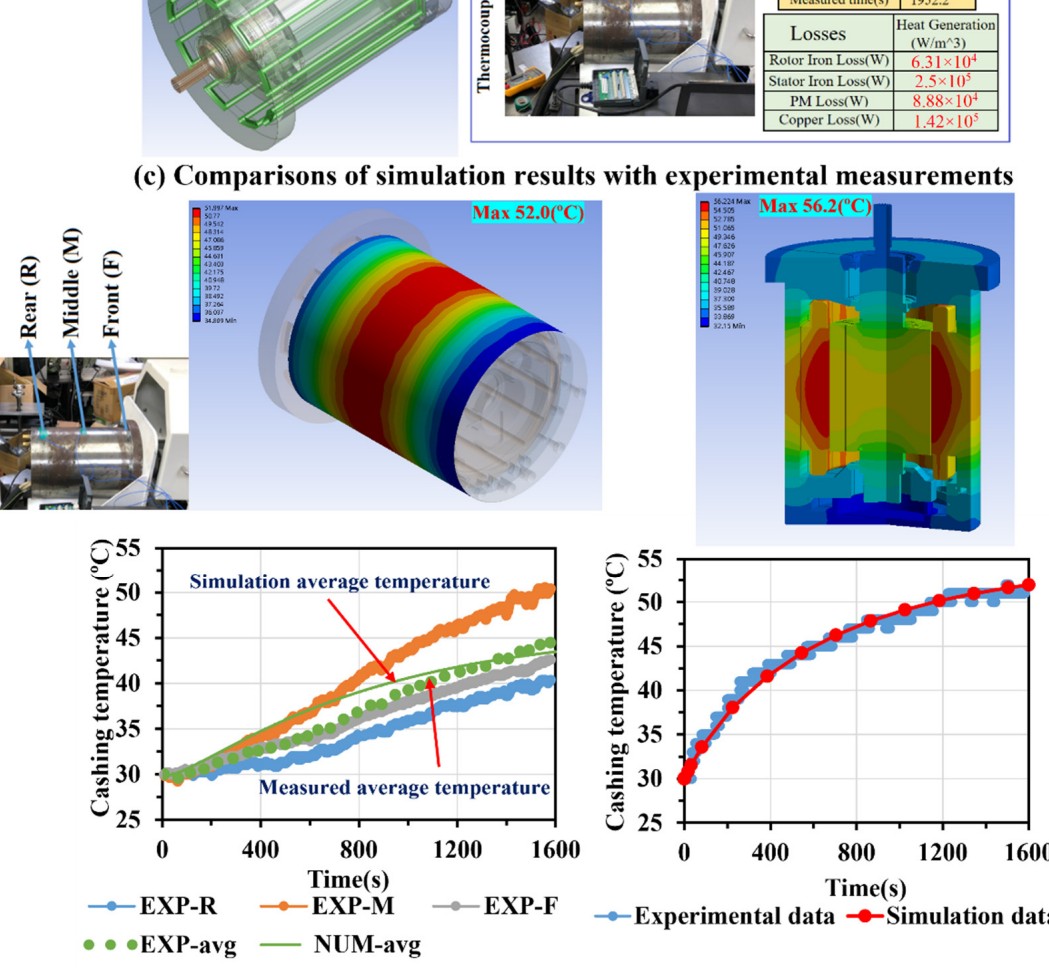

**Figure 8.** (**a**) configuration of electric motor simulated and tested (**b**) experimental rig and test condition for validation of simulation model (**c**) comparisons between simulation results and experimental measurements.

## 3. Results and Discussion

### 3.1. Experimental Results

For completeness, the experimental data reported in [39] to determine the $K_{eff}$ of the RTL with a filling ratio of 0.8 and the variation of $Nu_{ext,con}$ against $Re_\Omega$ defined by Equation (17) are collected in Figure 9a,b. In Figure 9a, the effective thermal conductivity of the RTL at a filling ratio of 0.8 is increased with $Q^*$ or Ca. In Figure 9b, the measured external Nusselt numbers ($Nu_{ext,con}$) of the rotating condenser bend for the various RTLs with different interior configurations and filling ratios collapse into a tight data trend that increases with $Re_\Omega$. Through the parametric analysis, the empirical $K_{eff}$ and $Nu_{ext,con}$ correlations are reported as Equations (20) and (21) [39], which are incorporated into the numerical model to calculate the effective thermal conductivity of the RTL in the rotor and the average heat-transfer rate ($h_{ext,con}$) of the rotating condenser bend in the air chamber.

$$K_{eff} = 1 + \left( 64.81 - 0.385Ca + 0.035Ca^2 \right) \times Q^{*(0.28 - 0.0005Ca + 3E-5Ca^2)} \quad \text{FR} = 0.8 \text{ with coil} \tag{20}$$

$$Nu_{ext,con} = 0.089 \times Re_\Omega^{0.59} \tag{21}$$

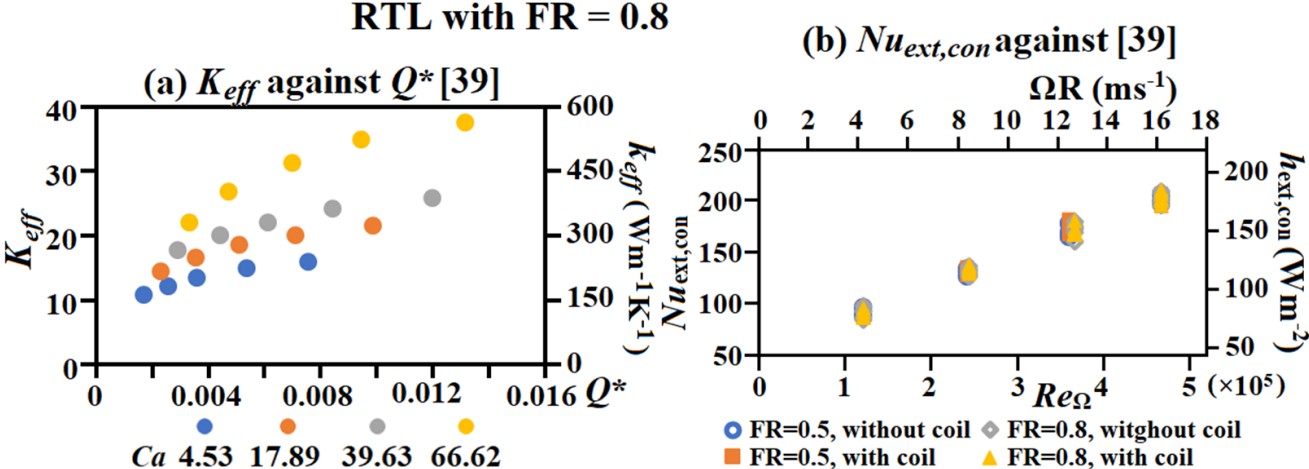

**Figure 9.** Variations of (**a**) $K_{eff}$ against $Q^*$ at various *Ca* (**b**) $Nu_{ext,con}$ against $Re_\Omega$ for the RTL with filling ratio of 0.8 [39].

The present transient thermography method is devised with the strategic aim of developing the $Nu_{inner,AC}$ correlation for evaluating the convective heat-transfer rate ($h_{inner,AC}$) over the inner annular surface of the air chamber with and without the RTLs. Figure 10 shows the variations of $Nu_{inner,AC}$ in the air chambers with and without the RTLs against $Re_\Omega$ that take the radius of the rotor as the characteristic length, while the characteristic length of $Nu_{inner,AC}$ is the diameter of the annular chamber. Even without the rotating condenser bend in the air chamber, the airflow is entrained by the rotating surfaces of the shaft and the disc-like end-facet of the rotor to raise $Nu_{inner,AC}$. As shown in Figure 10, all the $Nu_{inner,AC}$ values in the air chamber with and without the RTLs are increased with $Re_\Omega$. Enhanced by the stirring effect of the rotating condenser bend in the air chamber, the $Nu_{inner,AC}$ with RTL is consistently higher than that without RTL, as compared in Figure 10. Following the data trends driven by $Re_\Omega$ in Figure 10, the $Nu_{inner,AC}$ correlations are generated as Equations (15) and (16). The $h_{inner,AC}$ converted from the $Nu_{inner,AC}$ correlation, which is the function of rotor speed for the particular geometric configuration, is constantly used to define the convective thermal boundary condition in the motor to analyze the thermal field of the PMSEM.

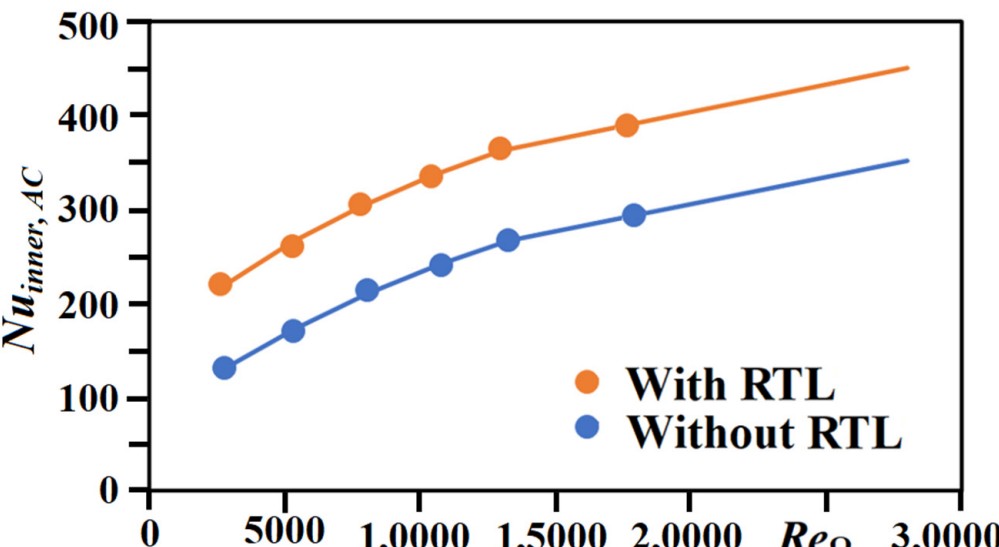

**Figure 10.** Variations of $Nu_{inner,AC}$ against $Re_\Omega$ in the air chambers with and without the RTLs.

*3.2. Numerical Results*

3.2.1. Baseline Condition

The baseline thermal fields of the PMSEMs with and without the RTLs at the $T_0$ operating condition with a rotor speed of 1200 rev/min and total electric current of 1000 A are compared in Figure 11. The comparisons of the temperature fields in Figure 11a–c reveal the thermal impacts of the RTLs. In Figure 11a without RTL, the ring-shaped high-temperature zone emerges at the mid-zone of the rotor to respond to the heat fluxes emitted from the magnets along the radially outer portion in the rotor. With the additional axial heat-transfer pathway along the single- or twin-end RTLs, the maximum temperature values and temperature gradients in the rotor are considerably moderated from those without RTL. With or without the RTL as an axial heat conductor, the rotor temperatures are decayed toward the two axial ends of the rotor from the central hot-zone in Figure 11. However, the improved heat flux transmissions in the rotor and in the front (rear) air chambers due to the presence of the RTLs reduce the required axial temperature gradients to transfer similar thermal power from the rotor to the air chambers, which will be demonstrated later. With the water jacket around the stator in which the twelve coiled windings are embedded, the maximum temperature on the inner annular surface of the stator is lower than that on the outer surface of the rotor. The large temperature drops from the outer surface of the rotor to the inner annulus of the stator across the rotor-to-stator air gap, where the Taylor–Couette vortices prevail, echo the weak airflow heat convection. The large convective thermal resistance of the Taylor–Couette vortices around the rotating rotor prohibits effective radial heat transmission from the rotor into the water flow in the cooling jacket through the stator. As the two axial ends of the extended steel frame of the stator are not in touch with the cylindrical water jacket, the hottest spot in the stator of each PMSEM emerges in the two axial ends of the stator steel frame. However, the stirring effect of the rotating RTL condenser-bends promotes the convective heat-transfer rate in the air chamber to enhance the convective heat-transfer from the hot ends of the stator steel frame into the airflow adjacent to the cold endplates of the motor casing. With a similar thermal load in the stator, the heat fluxes out of the two axial ends of the stator frame in the PMSEM with the RTLs are transferred at reduced wall-to-air temperature difference from that without RTL. The enhanced heat convection in the annular air chamber by the stirring effect of the RTLs has led to the reduced hot-spot temperature in the axial end of the stator frame.

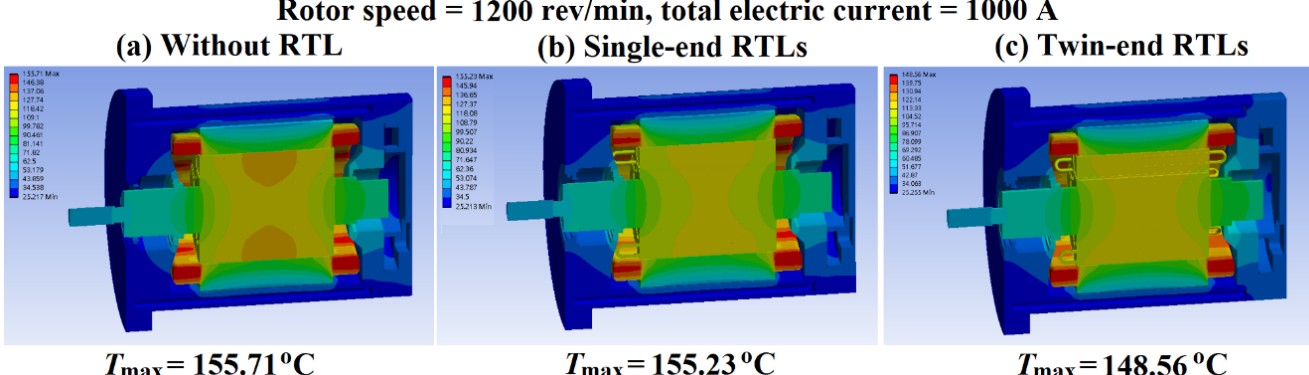

**Figure 11.** Detailed temperature distributions of the PMSEMs (**a**) without RTL and with (**b**) single-end RTLs; (**c**) twin-end RTLs at rotor speed of 1200 rev/min and total electric current of 1000 A.

Comparing the temperature distributions in the disc-type casing plates at the two axial ends of each PMSEM shown in Figure 11a–c, the warm zone in each end-casing plate adjacent to the air chamber with the RTLs is extended from that without RTL. Taking into account that the material conductive thermal resistance of the front/rear end-cover is identical for the PMSEMs with and without the RTLs, the augmented convective heat-transfer rate in the air chamber by the stirring effect of the RTLs increases the axial heat fluxes transferred into the end-casing plate. In other words, the axial and radial heat fluxes that are transferred from the axial facet of the rotor and the annular surface of the extended stator end-frame, respectively, are both promoted by the augmented airflow heat convection inside the air chamber due to the stirring effect of the rotating RTLs. The highest material temperature in the axial end of the stator end-frame is thus reduced from 155.71 °C in Figure 11a to 148.56 °C in Figure 11c. In Figure 11b with single-end RTLs, the material temperatures in the two hot axial ends of the stator end-frame are considerably different due to the different convective heat-transfer rates between the front and rear air chambers with and without the RTLs. Adjacent to the air chamber without RTL, the maximum temperature in the stator end-frame reaches 155.23 °C, similar to the hottest spot in the stator frame of the PMSEM without RTL. Nevertheless, the single-end RTLs still provide an effective heat-transfer pathway to axially transfer the heat flux out of the rotor into the front-end air chamber.

The improvement of rotor cooling performance in the PMSEMs by the RTLs at the baseline operating condition is demonstrated by Figure 12, in which the thermal fields of the rotor in the PMSEMs (a) without RTL and with (b) single-end RTLs and (c) twin-end RTLs are compared at the $T_0$ condition. In the rotor without RTL, the temperature distributions are basically symmetrical about the rotor axis. As the rotor is thermally entrapped by the surrounding Taylor–Couette vortices to impede the effective radial heat-transfer, the axial heat transmission becomes the crucial pathway for transferring the volumetric heat generations out of the rotor. The differential axial and radial thermal resistances for a rotor without RTLs result in the higher axial temperature gradients to transfer the heat flux. At the $T_0$ operating condition, the maximum temperature in the mid-way hot zone of the rotor without RTL reaches 131.85 °C in Figure 12a. The embodiment of the additional axial heat-transfer pathways (RTLs) in the rotor breaks the symmetrical temperature distributions about the rotor axis. In Figure 12b,c, the lower material temperatures adjacent to the RTLs of different lengths in the rotors create the asymmetrical angular and axial temperature fields. The maximum temperatures among the hot zones in the rotors with the single- and twin-end RTLs are reduced to 122.88 °C and 119.93 °C, respectively from 131.85 °C in Figure 12b,c. With the improved axial heat transmissions by the RTLs under the identical volumetric thermal power generated at $T_0$ condition, the temperatures at the end-facet of the rotor with the RTLs are higher than those without RTL. The RTLs promote the transfer of thermal power from the rotor core to the exposed condenser bends and elevate the

regional temperature gradients to augment the thermal diffusions around each RTL. The higher temperatures among the axial end-facet of the rotor with the RTLs further enlarge the wall-to-$T_g$ difference to increase the convective heat flux converting the interfacial conductive heat flux into the air chamber. Consequently, the heat flux directed toward the cold endplate of the motor casing through the air chamber is increased due to the increased $h_{inner,AC}$.

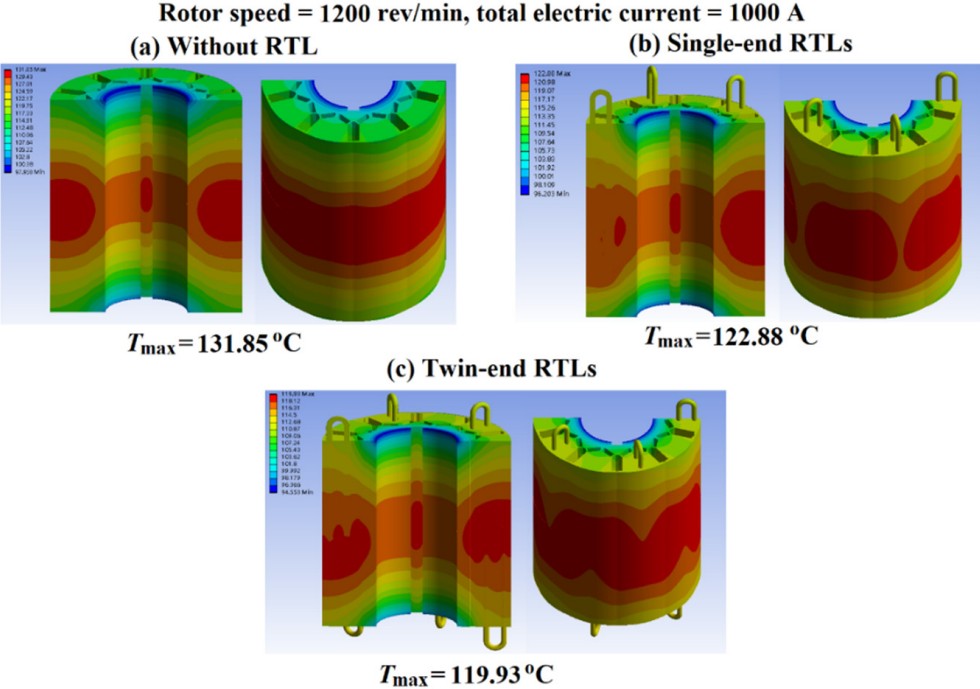

**Figure 12.** Thermal fields of the rotor in the PMSEMs (**a**) without RTL and with (**b**) single-end RTLs and (**c**) twin-end RTLs at $T_0$ operating condition.

While the presence of RTLs significantly affects the levels and distribution patterns of temperature in the rotor of the PMSEMs, the augmented $h_{ext,con}$ and $h_{inner,AC}$ for the airflow in the air chamber that is enclosed by the rotor end-facet, the annular steel frame of stator, the rotating shaft, and the cold end-cover of the motor casing affect the temperature gradients in the coiled windings, leading to its consequential influences on the thermal fields of the windings. Figure 13 depicts the temperature distributions of the coiled windings in the PMSEMs (a) without RTL and with (b) single-end RTLs and (c) twin-end RTLs at the $T_0$ operating condition. As compared in Figure 13, the patterns of temperature distribution of the coiled windings in the PMSEMs with and without the RTLs are similar. The hot zone is centered in the winding core with higher temperatures at the two axial ends in contact with the hot annular steel frame of the stator. The $T_{max}$ in each coiled winding remains higher than the maximum temperature in the rotor of each PMSEM owing to much higher copper loss than other losses, as shown in Table 2. The augmented $h_{inner,AC}$ in the air chamber due to the presence of the RTLs is beneficial for reducing the $T_{max}$ in the coiled windings, especially with the twin-end RTLs. In this respect, as indicated in Figure 13, the maximum temperatures in the core of the coiled windings for the PMSEMs without and with the single- and twin-end RTLs are 154.01 °C, 153.65 °C, and 147.11 °C, respectively.

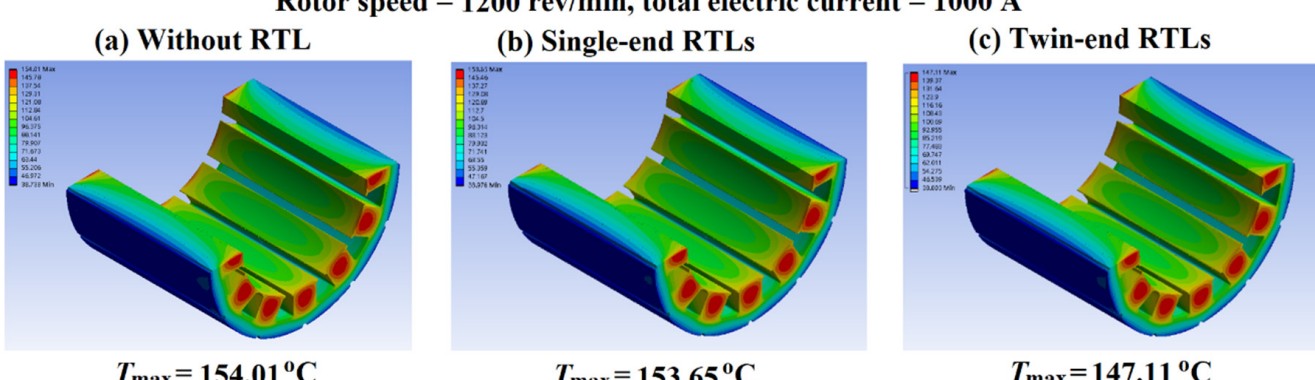

**Figure 13.** Detailed temperature distributions of the coiled windings in the PMSEM (**a**) without RTL and with (**b**) single-end RTLs and (**c**) twin-end RTLs at rotor speed of 1200 rev/min and total electric current of 1000 A.

The results compared in Figures 11–13 highlight the better thermal performance improvements by implanting the twin-end RTLs in the rotor. Nevertheless, as the temperatures in the rotor with the twin-end RTLs are reduced from those with the signal-end RTLs, the driven potential for motivating the phase-change activities in the RTL is accordingly moderated as a result of the reduced heat-source-to-sink temperature difference. Figure 14 compares the surface temperatures on the single- and twin-end RTLs at the $T_0$ operating condition. With the constant thermal load generated in the rotor, the respective ranges of surface temperatures on the single- and twin-end RTLs are 113.69–120.98 °C and 111.57–118.81 °C. Owing to the different interior configurations between the two axial ends of each PMSEM, as shown in Figure 6b, the cold-end condenser temperatures for the twin-end RTLs in the front and rear air chambers are different in Figure 14b. Based on these thermal conditions, the phase-change activities in the single- and twin-end RTLs at the different heat source-to-sink temperature differences result in the effective thermal conductivities of 3018.4 and 2251.7 $\mathrm{Wm^{-1}K^{-1}}$ with the attendant $T_g$ in the air chamber at 77.5 and 77 °C, respectively. Considering the subtle $k_{eff}$ difference between the single- and twin-end RTLs caused by different $Q^*$ at the identical $T_0$ operating condition, the differential overall temperature reductions in the rotor caused by the single- and twin-end RTLs have a negligible effect on their $k_{eff}$ values.

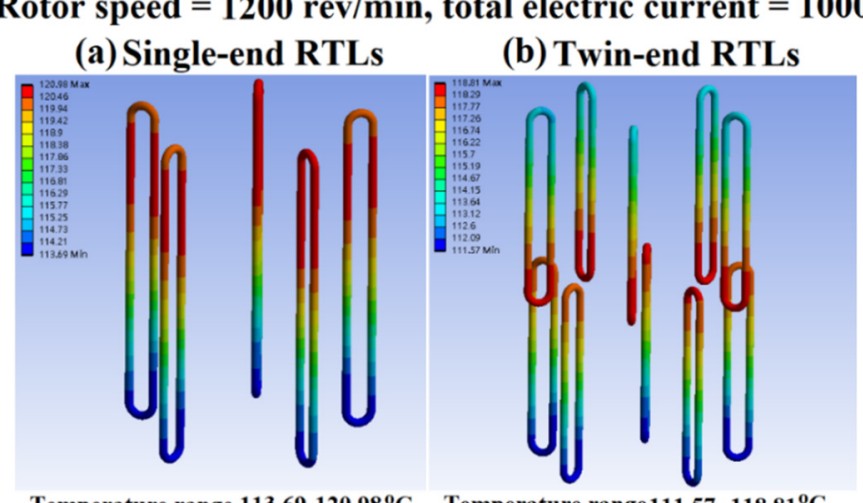

**Figure 14.** Surface temperatures on (**a**) single- (**b**) twin-end RTLs at $T_0$ operating condition.

### 3.2.2. $T_0$–$T_2$ Operating Conditions

The temperature maps of PMSEMs with and without RTLs at $T_0$ (a–c), $T_1$ (d–f), and $T_2$ (g–i) operating conditions are collected in Figure 15. As summarized in Table 2, the differential temperature maps between cases $T_0$ and $T_1$ in Figure 15 are produced by different rotor speeds of 1200 rev/min ($T_0$) and 1500 rev/min ($T_1$) with the identical total electric current (power input) for the PMSEMs. With the same rotor speed of 1200 rev/min at $T_0$ and $T_2$ conditions, the variations in the temperature maps from $T_0$ to $T_2$ cases are caused by raising the total electric current (electric power) fed into the motors. Notice that the color bars in all the figures are different, with the maximal value being automatically set to be the computed maximal temperature for each case. In general, the patterns of temperature distribution in the PMSEMs are characterized by the manner of RTL installation in the rotor, with negligible impact caused by the variation of rotor speed or total electric power input.

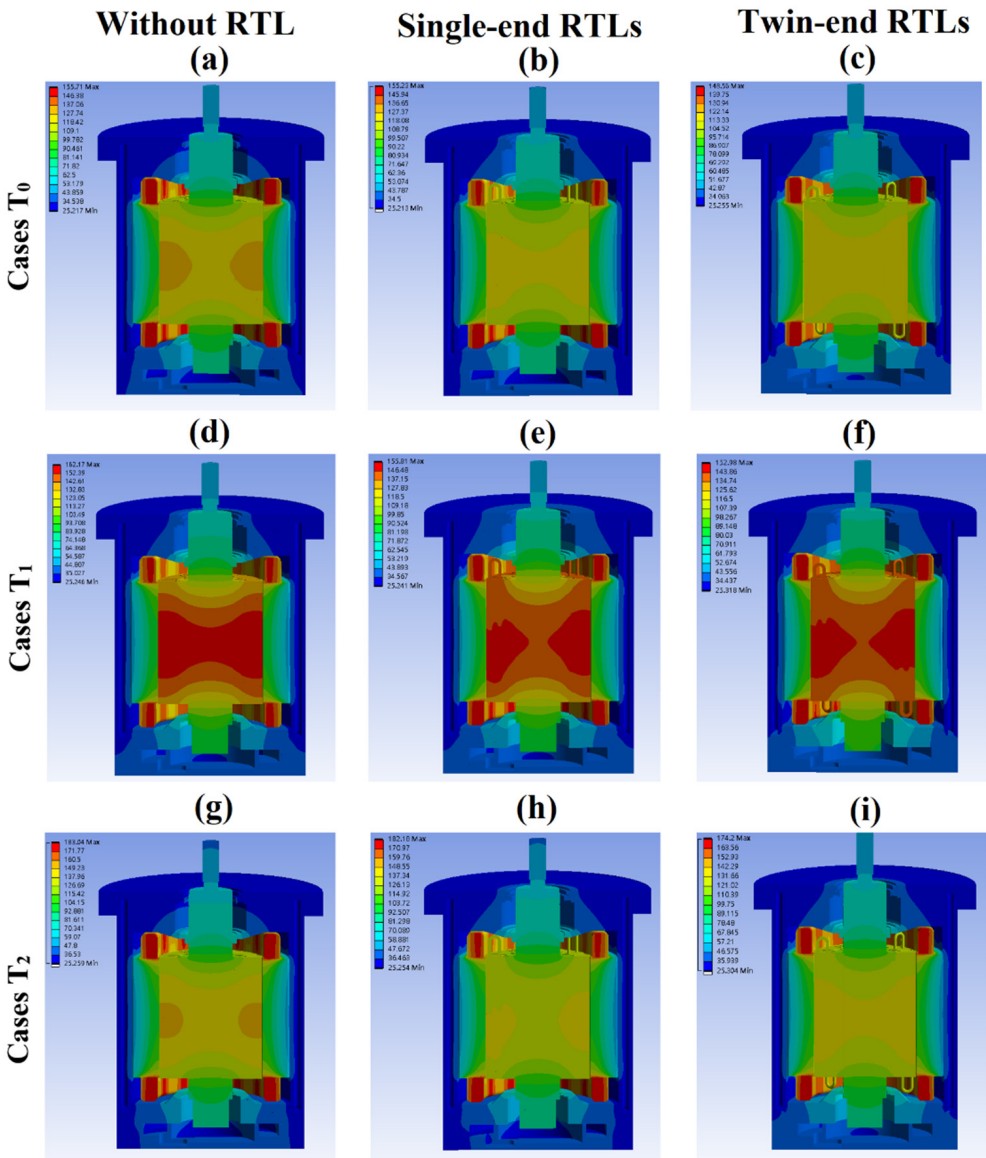

**Figure 15.** Temperature maps of PMSEMs with and without RTLs at $T_0$–$T_2$ operating conditions.

Cross examining the temperature maps obtained at $T_0$ and $T_1$ cases, the increase in rotor speed, with the attendant power-loss increases as compared in Table 2, considerably elevates the overall temperature levels in the PMSEM without RTL. As the values of $h_{ext,con}$ and $h_{inner,AC}$, as well as the effective thermal conductivities of the RTLs are increased with

rotor speed, the $T_{max}$ increases from $T_0$ (baseline) conditions at the $T_1$ case, as the $T_0$-to-$T_1$ upraised PM and iron losses for the PMSEMs with the single- or twin-end RTLs are considerably less than those without RTL. The elevated thermal loads owing to the drop of the electromagnetic efficiency by increasing the rotor speed from 1200 to 1500 rev/min considerably undermines the thermal performance for the PMSEM without RTL. When the operating condition is shifted from $T_0$ to $T_2$ at a rotor speed of 1200 rev/min to increase the total electrical current (input power) by 110%, the increases in the various power losses in Table 2 are less than those caused by changing the operation condition from $T_0$ to $T_1$, except for the copper loss. The comparison between the simulation results obtained at $T_0$ and $T_2$ conditions reveals more $T_2$ to $T_0$ temperature increases than the overall temperature increases by changing the operating condition from $T_0$ to $T_1$, in particular for the PMSEM without RTL. This result highlights the significance of the elevated $h_{ext,con}$ and $h_{inner,AC}$ by raising rotor speed on the thermal performance of the PMSEM. At the constant rotor speed, the $h_{ext,con}$ and $h_{inner,AC}$ remain similar. Although the $k_{eff}$ of the RTL is increased with $Q^*$ prior to its dry-out limit, the $T_2$ to $T_0$ temperature increases are still larger than the $T_1$ to $T_0$ counterparts for the PMSEMs with the single- and twin-end RTLs.

The quantitative assessment of the cooling performance improvements created by implanting the RTLs in the rotor is carried out by normalizing the maximum temperatures of motor ($T_{max}$), rotor (rotor $T_{max}$), stator (stator $T_{max}$), and shaft (shaft $T_{max}$) in the PMSEMs with the single- and twin-end RTLs against the $T_{max}$ references in the PMSEM without RTL. Table 3 summarizes the maximum temperature ratios of motor, rotor, stator, and shaft, as well as their maximum temperatures in the PMSEMs with and without the RTLs at $T_0$–$T_2$ conditions.

**Table 3.** Ratios of maximum temperatures of motor ($T_{max}$), rotor (rotor $T_{max}$), stator (stator $T_{max}$), and shaft (shaft $T_{max}$) in the PMSEMs with and without the RTLs at $T_0$–$T_2$ conditions.

| | $T_0$ Case | | | $T_1$ Case | | | $T_2$ Case | | |
|---|---|---|---|---|---|---|---|---|---|
| | Without RTL | Single RTL | Twin RTL | Without RTL | Single RTL | Twin RTL | Without RTL | Single RTL | Twin RTL |
| $T_{max}$ (°C) | 155.71 | 155.23 | 148.56 | 162.17 | 155.81 | 152.98 | 183.04 | 182.18 | 174.2 |
| $T_{max}$ ratio | 100.0% | 99.7% | 95.4% | 100.0% | 96.1% | 94.3% | 100.0% | 99.5% | 95.2% |
| rotor $T_{max}$ (°C) | 131.85 | 122.88 | 119.93 | 161.93 | 150.87 | 147.65 | 152.18 | 140.43 | 136.76 |
| rotor ratio | 100.0% | 93.2% | 91.0% | 100.0% | 93.2% | 91.2% | 100.0% | 92.3% | 89.9% |
| stator $T_{max}$ (°C) | 154.01 | 153.65 | 147.11 | 154.66 | 154.33 | 151.42 | 181.17 | 180.34 | 172.35 |
| stator ratio | 100.0% | 99.8% | 95.5% | 100.0% | 99.8% | 97.9% | 100.0% | 99.5% | 95.1% |
| shaft $T_{max}$ (°C) | 128.43 | 120.47 | 117.84 | 157.48 | 147.89 | 145.12 | 148.05 | 137.65 | 134.33 |
| shaft ratio | 100.0% | 93.8% | 91.8% | 100.0% | 93.9% | 92.2% | 100.0% | 93.0% | 90.7% |

As compared in Table 3, all the $T_{max}$ ratios in the PMSEM with the twin-end RTLs are less than the single-end RTLs counterparts. The $T_{max}$ ratios in rotor (shaft) for the PMSEMs with the twin-end RTLs are reduced to 91% (91.8), 91.2% (92.2), and 89.8% (90.7) at $T_0$, $T_1$ and $T_2$ conditions. In the motor (stator), the $T_{max}$ ratios in the PMSEMs with the twin-end RTLs at $T_0$, $T_1$ and $T_2$ conditions are 95.4% (95.5), 94.3% (97.9), and 95.2% (95.1). The thermal performance improvements by the RTLs in the rotor and shaft (rotating assemblies) are greater in general than those in the stator and the axial ends of the stator frame, where the maximum temperatures of the motor develop. At rotor speeds and electrical currents in the ranges of 1200–1500 rev/min and 1000–1200 A ($T_0$–$T_2$ operating conditions), the maximum temperatures in the rotors with the single- and twin-end RTLs are reduced 8–14 °C and 10–22 °C, respectively from those without RTL. In view of the comparative $T_{max}$ ratios between $T_1$ and $T_2$ cases in Table 3, the cooling performance improvements attributed to the RTLs in the rotating components are more effective when the thermal loads are raised by increasing the rotor speed owing to the attendant drop of electromagnetic

efficiency ($T_2$ case) rather than that in the $T_1$ case with the fixed rotor speed. By raising the speed of the RTLs, the increases in $h_{ext,con}$ and $h_{inner,AC}$ also play an important role for the thermal performance improvement generated by the RTLs.

Another thermal performance index is the temperature gradient that affects the thermal stress in an electric motor. The thermal gradient is converted from the heat flux field using Equation (22).

$$\frac{\partial T}{\partial x, y, z} = -\frac{q_{x,y,z}}{k_{x,y,z}} \tag{22}$$

Figure 16 depicts the distribution of axial heat flux ($q_x$) on the rotor surface in the PMSEMs (a) without RTL and with (b) single-end RTLs (c) twin-end RTLs at $T_0$ (baseline) operating condition. It should be mentioned that the origin of the coordinate system is set at the centerline and the mid-plane of the rotor. Therefore, in Figure 16, a positive $q_x$ value indicates heat transfer towards the top surface, while a negative $q_x$ value means heat transfer towards the bottom surface. Referring to Equation (22), the pattern of axial local heat flux in Figure 16 is regarded as the distributing manner of the axial temperature gradients on the rotor surface. As reflected by Figure 16a, the magnitudes of axial temperature gradient on the rotor surface are symmetrical about its mid-plane in the PMSEM without RTL. In Figure 16b, owing to the unilateral RTLs in the rotor, the axial temperature gradients are asymmetrical with the larger magnitudes among the end region without RTL. With the bilateral RTL arrangement in Figure 16c, the asymmetry in the axial distribution of $\partial T/\partial x$ is caused by the staggered arrangement of the RTLs in the two axial sides of the rotor. The magnitudes of $q_x$ and hence $\partial T/\partial x$ on the surface of the rotor with the twin-end RTLs are lower than that without RTL and with the single-end RTLs.

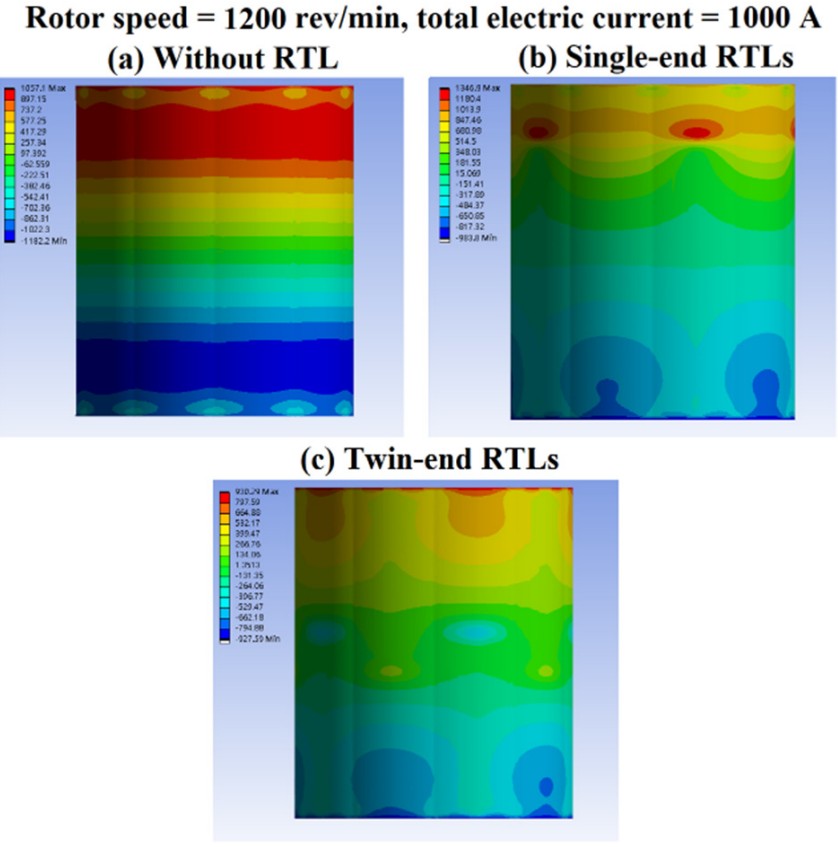

**Figure 16.** Distribution of axial heat flux ($q_x$) on the rotor surface in the PMSEMs (**a**) without RTL and with (**b**) single-end RTLs and (**c**) twin-end RTLs at $T_0$ operating conditions.

The average temperatures and axial temperature gradients of the rotor, the stator, and the shaft in the PMSEMs without and with the RTLs are summarized in Table 4.

**Table 4.** Averaged temperatures and axial temperature-gradients of PMSEMs at $T_0$ case.

| | Without RTL | Single-End RTLs | Twin-End RTLs |
|---|---|---|---|
| Rotor front surface $T_{avg}$ (°C) | 102.6 | 110.1 | 108.2 |
| Rotor back surface $T_{avg}$ (°C) | 114.2 | 110.0 | 110.3 |
| Rotor mid surface $T_{avg}$ (°C) | 130.8 | 121.7 | 118.8 |
| Rotor front temperature gradient (°C/m) | 373.5 | 152.6 | 140.5 |
| Rotor back temperature gradient (°C/m) | 219.6 | 154.0 | 112.5 |
| Stator front surface $T_{avg}$ (°C) | 111.3 | 102.0 | 101.5 |
| Stator back surface $T_{avg}$ (°C) | 104.5 | 107.1 | 103.4 |
| Stator mid surface $T_{avg}$ (°C) | 70.9 | 70.6 | 69.3 |
| Stator front temperature gradient (°C/m) | 535.4 | 416.0 | 427.0 |
| Stator back temperature gradient (°C/m) | 445.3 | 483.0 | 451.8 |
| Shaft front surface $T_{avg}$ (°C) | 96.3 | 94.5 | 92.9 |
| Shaft back surface $T_{avg}$ (°C) | 100.9 | 97.2 | 97.2 |
| Shaft mid surface $T_{avg}$ (°C) | 127.4 | 119.7 | 117.2 |
| Shaft front temperature gradient (°C/m) | 411.0 | 333.4 | 321.6 |
| Shaft back temperature gradient (°C/m) | 350.3 | 298.1 | 264.6 |

As indicated in Table 4, the presence of RTLs in the rotor considerably moderates the averaged axial temperature gradients in the rotor and shaft from those without RTL, especially for the twin-end RTLs. With the relatively low thermal conductivities of the silicon steel sheet in the rotor and the steel shaft to hinder the axial heat conductions out of these rotating components when the rotor is not equipped with an effective cooling scheme, the in-rotor RTLs promote the axial heat flux transmission, leading to the lower averaged axial temperature gradients in the rotor and the shaft. However, the unilateral thermosiphon implantation at the front end of the rotor reduces (increases) the averaged axial temperature gradient at the front (rear) end of the stator from that without RTL. While the RTLs in the rotor have a profound effect when it comes to reducing the axial temperature gradients in the rotor and the shaft from those without RTL, the effects of RTL on the radial and angular temperature gradients in the motor components are marginal. The considerable reductions in temperatures and axial temperature gradients in the rotating assemblies using the in-rotor RTLs assist with overcoming the thermal barrier that prohibits the increase in the power density of an electric motor by allowing further enhancement of the magnetic field in a rotor with the temperatures maintained at sustainable levels.

## 4. Conclusions

The embodiment of the rotating thermosyphon loops (RTLs) in a rotor is numerically demonstrated as an effective and efficient rotor-cooling method that improves the thermal performance of an electric motor. The effective thermal conductivity of the rotating thermosyphon loop and the convective thermal boundary conditions for simulating the temperature field of the permanent magnet synchronous electric motor (PMSEM), which are not available in the open literature, are measured using the steady-state thermocouple method [39] and the present transient thermography method. The thermal signatures of the PMSEMs without RTL and with the in-rotor single- and twin-end RTLs at the $T_0$, $T_1$–$T_2$ operating conditions specified in Table 2 are identified with the following salient points concluded.

1.  The effective axial heat-transfer pathway constructed by the RTL in the rotor acts synergistically with its stirring effect, which augments the convective heat-transfer in the air chamber to considerably reduce the temperatures in the rotating components

with moderate temperature reductions in the coiled windings of the stator and $T_{max}$ in the motor at $T_0$–$T_2$ operating conditions. At the rotor speeds and total electrical currents in the ranges of 1200–1500 rev/min and 1000–1200 A, the $T_{max}$ values in the rotors with the single- and twin-end RTLs are reduced 8–14 °C and 10–22 °C, respectively, from those without RTL.

2.  The effective thermal conductivity ($k_{eff}$) and the average convective heat-transfer coefficient on the rotating surface of the condenser bend ($h_{ext.con}$) for the RTL, as well as the heat-transfer rate on the annular surface of the front/rear air chamber with and without the RTLs increase with rotor speed. The higher degrees of thermal performance improvements attributed to the twin-end RTLs in the rotor emerge when the various losses in the PMSEM are increased by raising the rotor speed instead of adding the motor input power at a fixed rotor speed.

3.  The in-rotor twin-end RTLs considerably promote the axial heat flux transmission, leading to reduced axial temperature gradients of the rotating components from those without RTL. The combined reductions in temperatures and axial temperature gradients of the rotating assemblies by implanting the RTLs in a rotor permit the intensification of the magnetic flux of rotor with its temperatures at sustainable levels to assist in resolving the thermal barrier that hinders the further increase in power density of an electric motor.

**Author Contributions:** Conceptualization, S.W.C. and P.S.W.; methodology, S.W.C. and P.S.W.; software, P.S.W. and Y.E.L.; validation, Y.E.L.; experiments, W.L.C.; formal analysis and investigation, S.W.C., P.S.W. and M.-F.H., Y.E.L. and W.L.C.; resources, S.W.C., P.S.W. and M.-F.H.; data curation, Y.E.L. and W.L.C.; writing—original draft preparation, S.W.C.; writing—review and editing, S.W.C. and P.S.W.; supervision, S.W.C., P.S.W. and M.-F.H.; project administration, S.W.C., P.S.W. and M.-F.H.; funding acquisition, S.W.C. and M.-F.H. All authors have read and agreed to the published version of the manuscript.

**Funding:** This research was funded by Ministry of Science and Technology, Taiwan, R.O.C. under the grant numbers MOST 109-2622-8-006-005 and MOST 108-2221-E-006-090-MY3.

**Institutional Review Board Statement:** Not applicable.

**Informed Consent Statement:** Not applicable.

**Data Availability Statement:** Not applicable.

**Acknowledgments:** These authors acknowledge the evaluation of the power losses in the PMSEM by J. F. Caceres.

**Conflicts of Interest:** The authors declare no conflict of interest.

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
