# Peer review of "Thermal Performance Improvement by Rotating Thermosyphon Loop in Rotor of an Interior Permanent Magnet Synchronous Electric Motor"

_inventions, doi:10.3390/inventions7020037_

Round 1

Reviewer 1 Report

Main question addressed by the research: The work addresses the thermal Performance Improvement by Rotating Thermosyphon Loop in Rotor of an Interior Permanent Magnet Synchronous Electric Motor.
Originality and relevance of the topic: The topic is relevant to the field and it considers a suitable research gap.
Added value of the paper:  The manuscript takes into account the study of the thermal performance, however the main purpose of it is not clearly stated. The paper should include clearly why they are analysing several parameters and why they are needed. This should be added at the end of the introduction as the main goal of the paper highlighting the research gap as well.

Quality of figures: Quality of figures should be improved for readability such as Figure 6 and 8..
Formatting of equations should also be improved.

Author Response

Please see the attached PDF file. Thanks.

Reviewer 2 Report

1)  is there any uncertainity in the experimental work?

2)  ΩPage 6 not highlighted cross check 

3) There are two different D's I have seen in eq.3 as small "d" and eq.6 capital "D"is there any difference between among? authors are requested to  cross check once and use either capital or small d for diameter through the article 

4) eq.6  D2 is it diameter? if yes, what is that diameter refernce? if no then highlit please?

5) Page 10 the uncertainites are taken from other studies or calculated by the authors? 

6) please represent uncertainties in Tabular form it is easy to understand for the readers

7)  Authors requested to main the same units SI or CGS but not both through out the article, somewhere degrees are represented in  K and somewhere in 0C for Instance page in page 5, page 10, 

Author Response

Please see the attached pdf file. Thanks.
